ecology/microbiology/plant science

climate change, Costa Rica, endophytes, global warming, paramo, Poaceae

**Author for correspondence:**
Priscila Chaverri
e-mail: priscila.chaverriechandi@ucr.ac.cr

# Response of psychrophilic plant endosymbionts to experimental temperature increase

## Carolina Seas[1,2] and Priscila Chaverri[3,4]

[1]Vicerrectoría de Investigación, Laboratorio de Ecología Urbana, Universidad Estatal a Distancia (UNED), 2050 San José, Costa Rica
[2]Centro Agronómico Tropical de Investigación y Enseñanza (CATIE), Escuela de Posgrado, Turrialba, Costa Rica
[3]Escuela de Biología and Centro de Investigaciones en Productos Naturales (CIPRONA), Universidad de Costa Rica, 11501-2060, San José, Costa Rica
[4]Department of Plant Sciences and Landscape Architecture, University of Maryland, College Park, MD 20742, USA

PC, 0000-0002-8486-6033

Countless uncertainties remain regarding the effects of global warming on biodiversity, including the ability of organisms to adapt and how that will affect obligate symbiotic relationships. The present study aimed to determine the consequences of temperature increase in the adaptation of plant endosymbionts (endophytes) that grow better at low temperatures (psychrophilic). We isolated fungal endophytes from a high-elevation (paramo) endemic plant, *Chusquea subtessellata*. Initial growth curves were constructed at different temperatures (4–25°C). Next, experiments were carried out in which only the psychrophilic isolates were subjected to repeated increments in temperature. After the experiments, the final growth curves showed significantly slower growth than the initial curves, and some isolates even ceased to grow. While most studies suggest that the distribution of microorganisms will expand as temperatures increase because most of these organisms grow better at 25°C, the results from our experiments demonstrate that psychrophilic fungi were negatively affected by temperature increases. These outcomes raise questions concerning the potential adaptation of beneficial endosymbiotic fungi in the already threatened high-elevation ecosystems. Assessing the consequences of global warming at all trophic levels is urgent because many species on Earth depend on their microbial symbionts for survival.

# 1. Introduction

Recently, global climate change has become known as one of the major threats to biodiversity, causing both change and loss [1,2]. These changes are modifying species distributions and, thus, the interactions among organisms. A vast gap in knowledge exists regarding the consequences of climate change on symbiotic relationships [3] and, more specifically, those mutualistic or obligate interactions with microorganisms [4,5]. The available research shows that changes in microbial symbionts and microbiomes may have positive or negative effects on the host, which in consequence can severely alter ecosystem structure and function [3,6]. For example, a correlation has been shown between the incidence and severity of important microbial diseases in animals [7–10] and plants [11,12], leading in some cases to the extinction of the host species [13]. A study demonstrated that temperature negatively affected the association between bark beetles and their fungal mutualists [14]. Other studies established that increased ocean temperatures affected corals and their algal endosymbiont *Symbiodinium* but not endosymbiotic bacteria [15,16]. Additionally, other studies revealed that plants with mycorrhizal associations had better survival rates under drought conditions [17,18].

Another aspect of global warming effects on biodiversity is the ability of organisms to adapt or acclimate [19,20]. The State of the Fungi 2018 Report [21] mentions one key unresolved question: 'What is the relative importance of fungal adaptation, migration and acclimation?' Because climate change is occurring and will be challenging to stop or reverse, it becomes extremely relevant to understand how species will adapt to the changing climate. Empirical studies on this subject are scarce, especially for microbial symbionts. A few of the available experimental studies demonstrate that some mycorrhizal, soil and wood-decay fungi are able to adapt or even benefit from increased temperatures [22–24]. Instead, others report that fungi are adversely affected by warming. One report measured the growth of the fungal mycelium in soils at different temperatures in the Arctic area of Svalbard [25], revealing that some fungi are specifically psychrophilic/psychrotrophic and can only live in cold temperatures, between 10 and 20°C [26].

The present study aimed to determine the consequences of temperature increase in the growth and adaptation of plant endosymbionts (endophytes) that grow better at low temperatures (psychrophilic). Ideal ecosystems to study the impact of global warming on biodiversity are areas of extreme cold temperatures, for example, the high treeless plateaus of tropical Central and South America (= paramos). Research shows that changes of 1°C or 2°C in temperature can negatively affect species that are adapted to living only in these ecosystems [27]. It is predicted that climate change will displace ecosystem boundaries and drastically reduce the total size of tropical alpine areas [27]. Limited studies have characterized fungal species associated with plants from extreme-temperature ecosystems, such as the paramo [28–31]. To date, several psychrophilic fungal species have been studied [32,33], but little is known about their capacity to adapt and grow at temperatures outside their optimum range [25,34,35]. If plant endosymbionts (i.e. endophytes: *endo* = inside, *phyte* = plant) have been found in almost every plant species studied and many of them provide benefits to the plant [36–38], we hypothesize that the loss of the endosymbionts will lead to a cascade of negative effects on plant hosts and the fragile paramo ecosystem.

# 2. Materials and methods

## 2.1. Study system

The study was carried out in the paramo of Costa Rica, in the Talamanca mountain range, which covers 152 km$^2$ (15 205 ha). Specifically, we sampled in Cerro Buena Vista (3491 m, 9°33′11.16″ N, 83°45′24.26″ W) and Cerro Chirripó (3820 m, 9°29′2.7″ N, 83°29′19.2″ W). In this area, the temperature varies from 0 to 25°C, but during the dry season, it can reach −5°C and 28°C. The dry season lasts from November to April, and the rainy season, from May to October [39] (electronic supplementary material, figure S4). We isolated endophytic fungi from the endemic paramo species *Chusquea subtessellata* (Poaceae), which is considered one of the most abundant and easiest to identify species and is present in most of the paramo of Costa Rica [40]. *Chusquea subtessellata* is also a good colonizer of areas that have experienced scorches or fires and is important for the tapir's (*Tapirus bairdii*) diet [41,42].

## 2.2. Plant sampling and isolation of endophytic fungi

The literature indicates that *C. subtessellata* has a distribution range from 2200 to 3800 m [42]. However, in our study, plants could be found only starting at 3270 m. From this point to the highest point of each mountain

(i.e. 3500 m in Cerro Buena Vista and 3800 m in Cerro Chirripó), collecting was performed every 100 m of altitude, with collection locations separated by at least 50 m. Additionally, two plants (growth units) of *C. subtessellata* were identified at each point, where samples of leaves, stems and roots were taken. For Cerro Buena Vista, three collection points were selected every 100 m of altitude, for a total of 18 plants. In the case of Cerro Chirripó, six collection points were selected every 100 m of altitude, for a total of 56 plants (the total was not 72, since at 3300 and 3800 m plants were found at only two collection points).

Specific selection criteria for each plant sample were used to obtain consistent samples. The plants had to be without shade or with the least possible shade and without herbivory or other obvious damage [43–45]. Fungal endophytes were isolated from leaves, stems and roots. The leaves were collected 1 m from the ground, the stems were collected 50 cm from the ground, and the roots were excavated with a shovel at the base of the plant to take samples at a depth between 10 and 25 cm. The samples were stored in plastic bags and transported in coolers (without ice) to maintain a fresh environment for a period no longer than 24 h before processing.

The leaves, stems and rootlets collected from each plant were divided into three pieces of approximately $5 \times 5$ mm (e.g. 74 plants = 222 leaf pieces). To eliminate surface contaminants and epiphytes, the plant tissue pieces were subjected to sequential immersions in 2% sodium hypochlorite (1 min), 90% alcohol (a few seconds) and sterile distilled water (a few seconds) [25,31,46–48]. The surface-sterilized pieces were placed into individual Petri dishes with potato dextrose agar (PDA) supplemented with the antibiotic chloramphenicol to prevent the growth of endophytic or contaminating bacteria. The Petri dishes were incubated at a low temperature (15°C) to select the psychrophilic fungi until the mycelium began to grow from the plant tissue. The growing colonies were then transferred to Petri dishes with fresh PDA to obtain pure cultures. The isolation frequency, corresponding to the formula $Nd/Nt^{*}100$, was calculated, where $Nt$ is the total number of fragments and $Nd$ is the number of fragments in which endophytes were detected [49].

## 2.3. *In vitro* detection of psychrophilic fungi

Once the isolation process was completed, cultures were separated by morphotype, considering the site, altitude, plant part, coloration and macroscopic characteristics of the fungal colony. The morphotypes were grown in bioclimatic chambers with 12 h light/12 h dark at 10°C and 25°C to begin to discriminate the psychrophilic isolates. The morphotypes that showed the most radial growth at 10°C were considered psychrophilic and, therefore, used in subsequent analyses. In both cases, three replicates were used for each morphotype.

To determine if a fungal isolate was psychrophilic, we constructed initial growth curves ($GC_0$). The selected morphotypes were grown in Petri dishes with fresh PDA at four temperatures, 4, 10, 20 and 25°C, in a bioclimatic chamber with 12 h light/12 h dark. Radial growth (mm) was measured until 192 h. When obtaining the growth curves, an isolate was confirmed as being psychrophilic when its optimum growth temperature ($OT_0$) was between 10 and 20°C [26]. Four replicates were examined, and the use of the chambers was randomized to reduce the experimental error. The experiment followed a Latin square design with subdivided plots (electronic supplementary material, figure S5).

## 2.4. *In vitro* growth response at increased temperatures

### 2.4.1. Experiment #1

After calculation of the $GC_0$, the psychrophilic isolates were subjected to repeated increases in temperature to determine their ability to adapt to thermal variation. All the isolates were subjected repeatedly (for 10 cycles) to a temperature 5°C higher than their optimum growth temperature ($OT_0 + 5°C$) (experiment #1). For example, if the $OT_0$ of a fungal isolate was 15°C, the isolate was subsequently incubated and subjected to 10 cycles at a temperature of 20°C. Specifically, a plug collected from the edge of a freshly growing colony was placed into a Petri plate containing PDA. The plate was incubated in the dark at $OT_0 + 5°C$ for 10 days and replicated four times (first cycle). After 10 days, a new plug from this new colony (second cycle) was placed into a new Petri plate and incubated again at $OT_0 + 5°C$. This procedure was repeated until 10 cycles were completed.

### 2.4.2. Experiment #2

A second experiment was carried out to determine the response of the fungi to temperature variations in short periods, imitating the natural daily temperature variations in the paramo. The procedure

described previously for experiment #1 was followed, but the incubation temperature was not constant; instead, it alternated between 10°C and 20°C at 12 h intervals (day and night). After the 10 cycles in both experiments (#1 and #2), the growth curves were constructed again (final growth curves) using four temperatures, 4, 10, 20 and 25°C, in a bioclimatic chamber with 12 h light (day) / 12 h darkness (night) (electronic supplementary material, figure S6).

## 2.5. Data analyses

Quadratic curves were adjusted for each putative species per site, altitude, plant part and replicate; then, with the parameters of each (B0, B1, and B2), the growth curves ($GC_0$, #1 and #2) were grouped with a cluster analysis based on a two-way analysis of variance (ANOVA) implemented in the vegan R package [50]. The behaviour of $GC_0$ from experiments #1 and #2 and final growth curves was evaluated by generalized additive models with putative species as a random effect. The models were fit using the mgcv R package [51] with the Tweedie distribution function. The graphs of the curves were made using the R graph gallery.

## 2.6. Identification of the psychrophilic endophytes

The endophytic psychrophilic fungi were identified using the barcode for fungi, i.e. the internal transcribed spacer (ITS) of nuclear ribosomal DNA [52]. DNA was extracted using a Prepman™ Ultra Reagent (Thermo Fisher Scientific, New York, USA) commercial kit, and polymerase chain reaction (PCR) was performed with the ITS primers 5 (forward: GGAAGTAAAAGTCGTAACAAGG) and 4 (reverse: TCCTCCGCTTATTGATATGC) [53], which include the ITS1, 5.8S and ITS2 regions. Purification and sequencing of the PCR products were performed at Macrogen (Maryland, USA). The assembly and alignment of sequences were performed using Geneious 10 [54]. ITS sequences were then clustered into putative species or operational taxonomic units (OTUs) using the farthest neighbour algorithm implemented in Geneious 10 [54]. Sequences were assigned to the same OTU if their similarity was 99% or more, as suggested in previous studies [55,56]. Isolates were identified using the BLAST algorithm implemented in GenBank (www.ncbi.nlm.nih.gov/BLAST). The average sequence length submitted as a query was 500–600 bp, and the minimum sequence coverage was set to 90%. Resulting sequences are publicly available in Genbank under submission numbers MT882123–MT882198.

# 3. Results

We isolated fungal endophytes from the endemic paramo plant *Chusquea subtessellata* (Poaceae) in Costa Rica. A total of 74 plants were sampled (leaves, stems and roots), and 9% of their endophytes were psychrophilic (electronic supplementary material, table S1). The cluster analysis classified the psychrophilic isolates into three groups based on their initial growth curves with ANOVA (electronic supplementary material, figure S1). The isolates in the first group (G1) grew best between 15 and 20°C; the growth was greater at these temperatures than at 4°C, and at 25°C, the growth was greater than at 4°C but lower than the average growth between 15 and 20°C. The second group (G2) generally grew very little (on average less than 10 mm over the entire temperature range). The growth of the G2 group occurred only between 10 and 20°C, with maximum growth observed at 15°C. The third group (G3) on average grew less than 23 mm in the 8-day period; the maximum growth occurred at approximately 15°C, and growth was greater between 10 and 20°C than at 25°C, but at the same time, the average growth at 25°C was greater than that observed at 4°C. The initial growth curves were constructed according to the average radial growth (mm) obtained at 4, 10, 20 and 25°C for the putative species (figure 1) (electronic supplementary material, figure S2 shows the adjustment of the growth curves obtained for these three groups: $R^2 = 0.40$, $p < 0.05$).

After the two experiments with 10 cycles of repeated increases in temperature from the optimum ($OT_0 + 5°C$), the final growth curves indicated slower growth than the initial curves, even when the adaptive responses varied. The final growth curves were adjusted to four temperatures (4, 10, 20 and 25°C) for the three groups after both experiments ($R^2 = 0.368$, $p > 0.05$) for comparison with the initial growth curves. Some growth was always observed over time during the 10 cycles; however, in many cases, this growth was very weak (figure 1). The growth behaviour varied under the $OT_0 + 5°C$ increase (experiment #1) and temperature changes between day and night (experiment #2) ($R^2 = 0.497$,

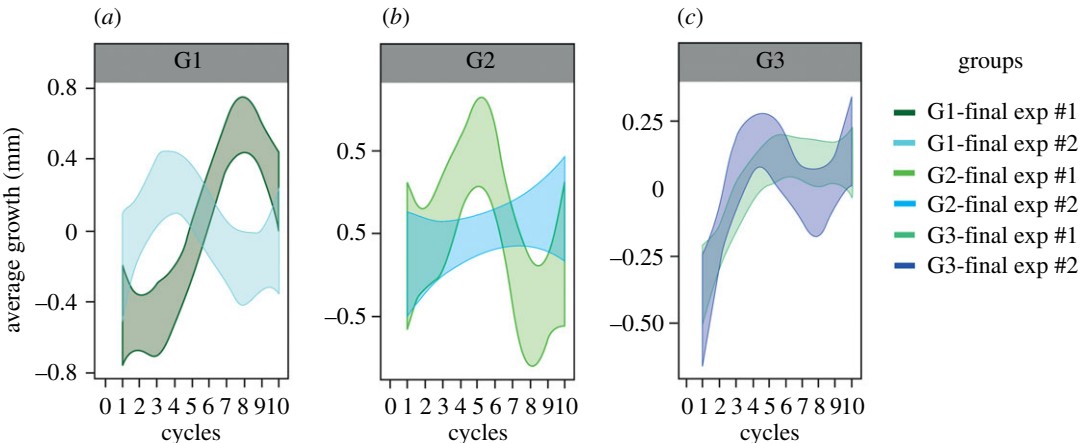

**Figure 1.** Average growth (mm) of the psychrophilic fungi groups (G1, G2, G3) during the 10 cycles of $OT_0 + 5°C$ increase (experiment #1) and temperature change between day (20°C) and night (10°C) (experiment #2), after 10 days. $OT_0 + 5°C =$ optimum temperature for the isolate + 5°C.

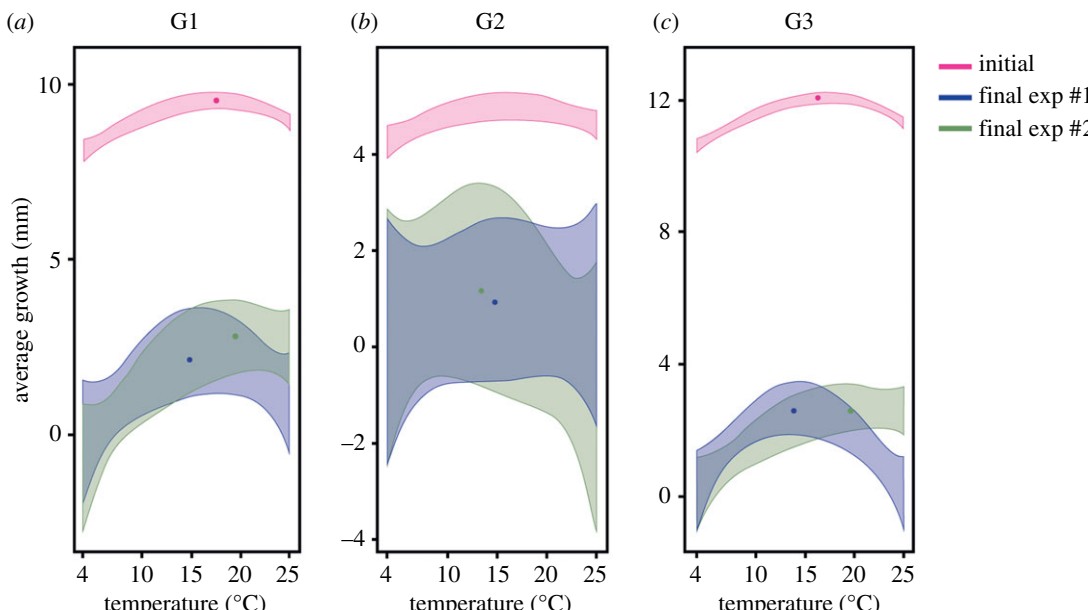

**Figure 2.** Comparison of initial and final growth curves according to each group (G1, G2, G3). initial = initial growth curve (GC$_0$); final exp #1 = final growth curve after experiment #1, 10 cycles of $OT_0 + 5°C$ increase; final exp #2 = final growth curve after experiment #2, 10 cycles of $OT_0 + 5°C$ increase between day (20°C) and night (10°C). Dots (•) on each curve represent the mean growth of each curve.

$p = 0.00494$). The final growth curves showed that the isolates in Group 1 (G1) grew more slowly than indicated by their initial curves but were able to keep growing well at temperatures higher than 20°C (figure 2a). For G2, slower growth, but with a large variance, was observed (figure 2b). In addition, this seems to indicate that the isolates were able to grow slightly better at lower temperatures than before the experiments. In G3, the shape of the curve was maintained, but the growth averages were much slower than the initial values (figure 2c). G3 seems to have had the most impact on growth.

Adaptative responses to the treatments were seen in specific examples (figure 3): *Arthrinium serenense* (G1) did not survive the increase in temperature. *Paracamarosporium* sp. (G1) did not endure the $OT_0 +$ 5°C increase (experiment #1) but did survive the temperature change between day and night (experiment #2). *Microdochium lycopodinum* (G2) maintained its growth behaviour, but its average radial growth decreased. Finally, *Trichoderma* cf. *asperellum* (G3) modified its behaviour and even ceased to be psychrophilic after the temperature increase experiment.

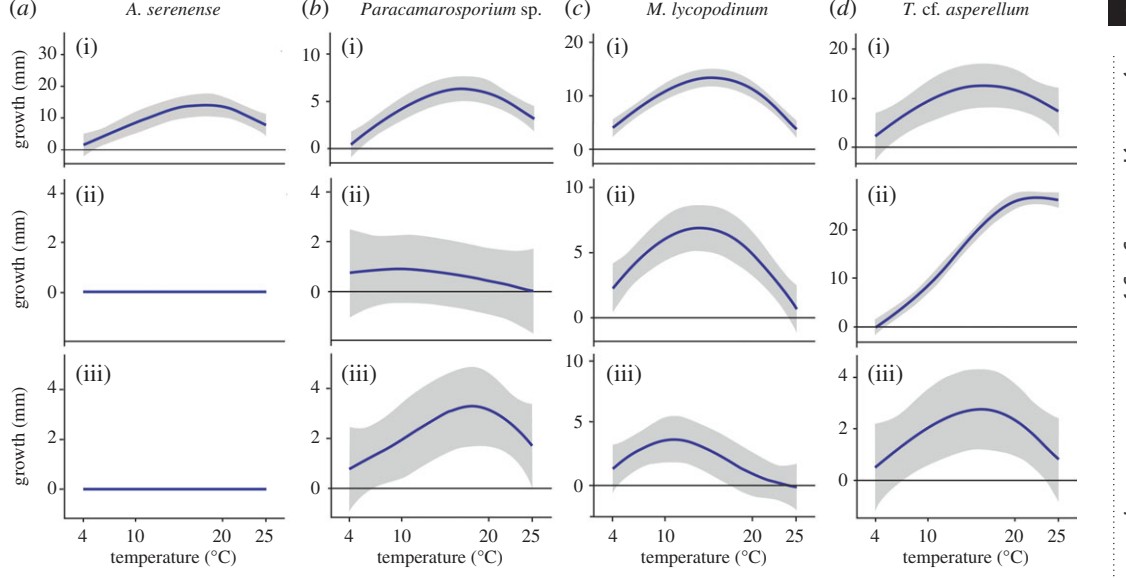

**Figure 3.** Examples of four psychrophilic fungi with varying responses to 10 cycles of $OT_0 + 5°C$ increase (experiment #1) (row 2) and temperature change between day (20°C) and night (10°C) (experiment #2) (row 3). $OT_0 + 5°C$ = optimum temperature for the isolate + 5°C. Row 1 is the initial growth curve. Y axis = growth in mm, X axis = temperature (4–25°C). Grey area represents standard deviation. Graphs of all isolates studied are in electronic supplementary material, figure S3.

## 4. Discussion

The fact that the experimental design included more than one sampling site and, additionally, an altitudinal gradient, was reflected in the fact that the frequency of isolation was greater than that reported in other studies of endophytic fungi from ecosystems with extreme cold temperatures and from living tissue of Poaceae and other families [25,49,57]. However, these values were similar to the values obtained in studies with more than one sampling site or those focused on paramo ecosystems, although not specifically the Poaceae [29,31].

This study demonstrates that most psychrophilic fungi isolated from the paramo endemic plant *Chusquea subtessellata* are negatively affected by short periods of increased temperatures. The responses vary, where some fungi die, others grow slower and decrease their optimum growth temperature, and others seem to adapt better and even increase their optimum growth temperature, ceasing to be psychrophilic. Previous studies have demonstrated that there is a high specificity of extreme-temperature endophytes with their host [49,58–61]. With this information, it is now possible to infer that many endosymbiotic fungal species and functional groups may be impaired or may disappear with global warming.

If all the plants in a natural ecosystem are in symbiosis with fungi, then changes in the endophytic community could have serious consequences for plant health, plant community structure and ecosystems [62,63]. The effects on the psychrophilic fungi studied here are not only manifested at the taxonomic level but also in terms of ecological functions. For example, *Arthrinium* is usually found in several bamboo species and has hypothesized functions, such as roles in heat and cold tolerance, production of anti-fungal and anti-herbivory substances, plant saprotrophy and plant disease [61,64–67]. *Trichoderma* and *Purpureocillium* are well-known genera with anti-fungal and anti-insect properties, respectively [37,68,69]. Some *Trichoderma* species are also plant-growth promoters and protect plants against drought or extreme conditions [68,70–72]. Some species of *Paracamarosporium* (previously known as *Camarosporium*) are halotolerant [73–76]. Therefore, the loss of the fungal symbiont and its ecological functions may contribute to a lower fitness of the host and adaptability to its environment and subsequent extinction [62,63].

The mechanisms by which fungi acclimate to low or high temperatures can be explained by adaptations in their physiology. Heat-shock or cold-shock responses are also known to occur in microorganisms, where a significant increase or decrease in ambient temperature starts stress responses that, in many cases, can be lethal to the organism or its host, or hinder their interactions [77,78]. For example, Arthur & Watson [34] showed a direct correlation between the growth temperature and the degree of lipid saturation in the

membrane. Therefore, it is expected that the cell membranes of psychrophilic fungi have fewer lipids, more cold-active enzymes, compatible solutes, and intracellular trehalose and greater synthesis of melanin and cyclosporine [33]. Increased temperatures could then lead to changes in the membrane composition affecting the ability of fungi to grow over specific temperature ranges [35]. More recently, Crowther & Bradford [22] suggested that the efficiency of growth at different temperatures is due to evolutionary compensations in the structures of the enzymes and cell membranes associated with the biochemical adaptation to temperature.

Although fungi and other microbes may benefit from increases in temperature, many regions, such as alpine, arctic or paramo areas, and their biodiversity continue to be the most affected by climate change [27,79,80]. Now, it is crucial to continue with long-term studies that will allow us to determine whether the adaptation or maladaptation to changes in temperature by these fungi, as expected to occur with the effects of climate change, can impact ecological networks and ecosystems.

Data accessibility. All DNA sequences are publicly available in GenBank MT882123–MT882198.

Authors' contributions. P.C. conceived the study, wrote the paper and supervised the graduate student C.S. C.S. performed the field and laboratory work, analysed data and wrote the paper.

Competing interests. On behalf of all authors, the corresponding author states that there is no conflict of interest.

Funding. This study was supported by AMI-UNED and Vicerrectoría de Investigación, Universidad de Costa Rica (B7048).

Acknowledgements. Thanks to Zaidett Barrientos and Julián Monge for support and Bryan Finnegan and Sergio Vílchez for guidance. Special thanks to all the collaborators during our field and laboratory work: Cristhian Ureña, Andrea Orellana, Grace Cobos, Beatriz Segura, Luis Pedro Utrera, Andrea Paíz, Andrea Pacheco, Maricela Pizarro, Maribel Zúñiga, Xaviera Amador. Finally, we are thankful to the project AMI-UNED 'Neotropical paramos: Threats to the ecosystem from global change' and the UCR project 'Effects of temperature changes in the endophytic fungi of paramo plants in Costa Rica' for funding.

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
