## [Reviewer comments · Royal Society Open Science]

Review History

RSOS-201405.R0 (Original submission)

Review form: Reviewer 1

Is the manuscript scientifically sound in its present form?

Yes

Are the interpretations and conclusions justified by the results?

No

Is the language acceptable?

Yes

Do you have any ethical concerns with this paper?

No

Have you any concerns about statistical analyses in this paper?

No

Recommendation?

Accept with minor revision (please list in comments)

Comments to the Author(s)

The manuscript entitled on "Response of psychrophilic plant endosymbionts to experimental temperature increase." Authors investigated about the psychrophilic fungi which were negatively affected by temperature increases and pointed out about assessing the consequences of global warming at all trophic levels is urgent because many species on Earth depend on their microbial symbionts for survival. Article is very well written and answered the issues raised by the other reviewer.

I have few queries before the acceptance.

Minor issues:

1. In abstract authors should mention about which plant have been used for the isolation of psychrophilic endosymbionts.
2. Authors should give Rationale for selecting the particular plant species and significance of this study in present context except climate change.
3. That would be better for the readers if authors can give the primer sequences.
4. Please check the text for spelling errors [such as alpine, artic or paramo areas]
5. Conclusion may be added in the last to clear the outcomes of this work.

Decision letter (RSOS-201405.R0)

Dear Dr Chaverri

On behalf of the Editors, we are pleased to inform you that your Manuscript RSOS-201405 "Response of psychrophilic plant endosymbionts to experimental temperature increase" has been accepted for publication in Royal Society Open Science subject to minor revision in accordance with the referees' reports. Please find the referees' comments along with any feedback from the Editors below my signature.

Please submit your revised manuscript and required files (see below) no later than 7 days from today's (ie 27-Oct-2020) date. Note: the ScholarOne system will 'lock' if submission of the revision is attempted 7 or more days after the deadline. If you do not think you will be able to meet this deadline please contact the editorial office immediately.

on behalf of Dr Yhasmin Mendes de Moura (Associate Editor) and Pete Smith (Subject Editor)
openscience@royalsociety.org

Associate Editor Comments to Author (Dr Yhasmin Mendes de Moura):

Associate Editor: 1

Comments to the Author:

Dear authors,

I have seen that you have addressed the major concerns from the reviewers and have seen the changes implemented in the manuscript, therefore, improving the overall scientific contribution of your study. I recommend this manuscript for acceptance with minor revision.

Thank you for your contribution for Royal Society Open Science and we hope to receive more scientific contributions from you in the future.

Best regards,
Yhasmin M Moura

Associate Editor: 2

Comments to the Author:

Dear authors,

The paper entitled: "Response of psychrophilic plant endosymbionts to experimental temperature increase" was assigned to evaluation. It is a very specific topic, but I consider a very interesting and with important results due to lack of knowledge on the impacts of climate change on endophyte symbionts of plants.

I would recommend this paper to go to peer review.

Sincerely,

Reviewer comments to Author:

Reviewer: 1

Comments to the Author(s)

The manuscript entitled on "Response of psychrophilic plant endosymbionts to experimental temperature increase." Authors investigated about the psychrophilic fungi which were negatively affected by temperature increases and pointed out about assessing the consequences of global warming at all trophic levels is urgent because many species on Earth depend on their microbial symbionts for survival. Article is very well written and answered the issues raised by the other reviewer.

I have few queries before the acceptance.

Minor issues:

1. In abstract authors should mention about which plant have been used for the isolation of psychrophilic endosymbionts.
2. Authors should give Rationale for selecting the particular plant species and significance of this study in present context except climate change.
3. That would be better for the readers if authors can give the primer sequences.
4. Please check the text for spelling errors [such as alpine, artic or paramo areas]
5. Conclusion may be added in the last to clear the outcomes of this work.

===PREPARING YOUR MANUSCRIPT===

===PREPARING YOUR REVISION IN SCHOLARONE===

Author's Response to Decision Letter for (RSOS-201405.R0)

See Appendix A.

Decision letter (RSOS-201405.R1)

Dear Dr Chaverri,

It is a pleasure to accept your manuscript entitled "Response of psychrophilic plant endosymbionts to experimental temperature increase" in its current form for publication in Royal Society Open Science.

You can expect to receive a proof of your article in the near future. Please contact the editorial office (openscience_proofs@royalsociety.org) and the production office (openscience@royalsociety.org) to let us know if you are likely to be away from e-mail contact -- if

you are going to be away, please nominate a co-author (if available) to manage the proofing process, and ensure they are copied into your email to the journal.

on behalf of Dr Yhasmin Mendes de Moura (Associate Editor) and Pete Smith (Subject Editor)
openscience@royalsociety.org

Appendix A

Response to reviewers:

1. In abstract authors should mention about which plant have been used for the isolation of psychrophilic endosymbionts.

R/ We have added this in the abstract.

2. Authors should give Rationale for selecting the particular plant species and significance of this study in present context except climate change.

R/ This is already mentioned in lines 94-99 of the Methods section.

3. That would be better for the readers if authors can give the primer sequences.

R/ We have added primers in the Methods section.

4. Please check the text for spelling errors [such as alpine, artic or paramo areas]

R/ Fixed.

5. Conclusion may be added in the last to clear the outcomes of this work.

R/ We would prefer not to add a Conclusions section as to avoid repetition. In the abstract is mentioned the main outcomes of this work (lines 26-29). They are also mentioned in the Discussion (paragraph starting on line 243).